# Harnessing Natural Polymers for Nano-Scaffolds in Bone Tissue Engineering: A Comprehensive Overview of Bone Disease Treatment

**Sushmita Saurav** [1], **Prashish Sharma** [1], **Anil Kumar** [2], **Zeba Tabassum** [1], **Madhuri Girdhar** [3], **Narsimha Mamidi** [4,*] and **Anand Mohan** [1,*]

1   School of Bioengineering and Biosciences, Lovely Professional University, Phagwara 144401, Punjab, India; sushmitasauravhanuman@gmail.com (S.S.); prashish.sharma15@gmail.com (P.S.); zebatabassum326@gmail.com (Z.T.)
2   Gene Regulation Laboratory, National Institute of Immunology, New Delhi 110067, Delhi, India; anilk@nii.ac.in
3   Division of Research and Development, Lovely Professional University, Phagwara 144401, Punjab, India; madhurigirdhar007@gmail.com
4   Wisconsin Centre for Nano Biosystems, School of Pharmacy, University of Wisconsin-Madison, Madison, WI 53705, USA
*   Correspondence: nmamidi@wisc.edu (N.M.); anand.20770@lpu.co.in (A.M.)

**Abstract:** Numerous surgeries are carried out to replace tissues that have been harmed by an illness or an accident. Due to various surgical interventions and the requirement of bone substitutes, the emerging field of bone tissue engineering attempts to repair damaged tissues with the help of scaffolds. These scaffolds act as template for bone regeneration by controlling the development of new cells. For the creation of functional tissues and organs, there are three elements of bone tissue engineering that play very crucial role: cells, signals and scaffolds. For the achievement of these aims, various types of natural polymers, like chitosan, chitin, cellulose, albumin and silk fibroin, have been used for the preparation of scaffolds. Scaffolds produced from natural polymers have many advantages: they are less immunogenic as well as being biodegradable, biocompatible, non-toxic and cost effective. The hierarchal structure of bone, from microscale to nanoscale, is mostly made up of organic and inorganic components like nanohydroxyapatite and collagen components. This review paper summarizes the knowledge and updates the information about the use of natural polymers for the preparation of scaffolds, with their application in recent research trends and development in the area of bone tissue engineering (BTE). The article extensively explores the related research to analyze the advancement of nanotechnology for the treatment of bone-related diseases and bone repair.

**Keywords:** tissue engineering; nano-scaffolds; biopolymers; nanostructure; bone regeneration

## 1. Introduction

The emerging area of bone tissue engineering employs the principles of bioengineering and life sciences to build biomaterials through the utilization of natural polymers that promote bone regeneration. In recent decades, the application of nanotechnology in regenerative medicine has developed significantly. The interconnection between cells and biomaterials at the nanoscale has escalated interest in the realm of nanotechnology as a tool for regenerative medicine. Studies have shown that materials with nanostructures or those that affect the nanostructure of the cellular microenvironment have advantages over the microscale equivalents [1]. The provenance of natural polymers ranging from the macroscale to the nanoscale, in which polysaccharides, nucleic acids and protein are applied as nanomaterials, with their different application are shown in Figure 1.

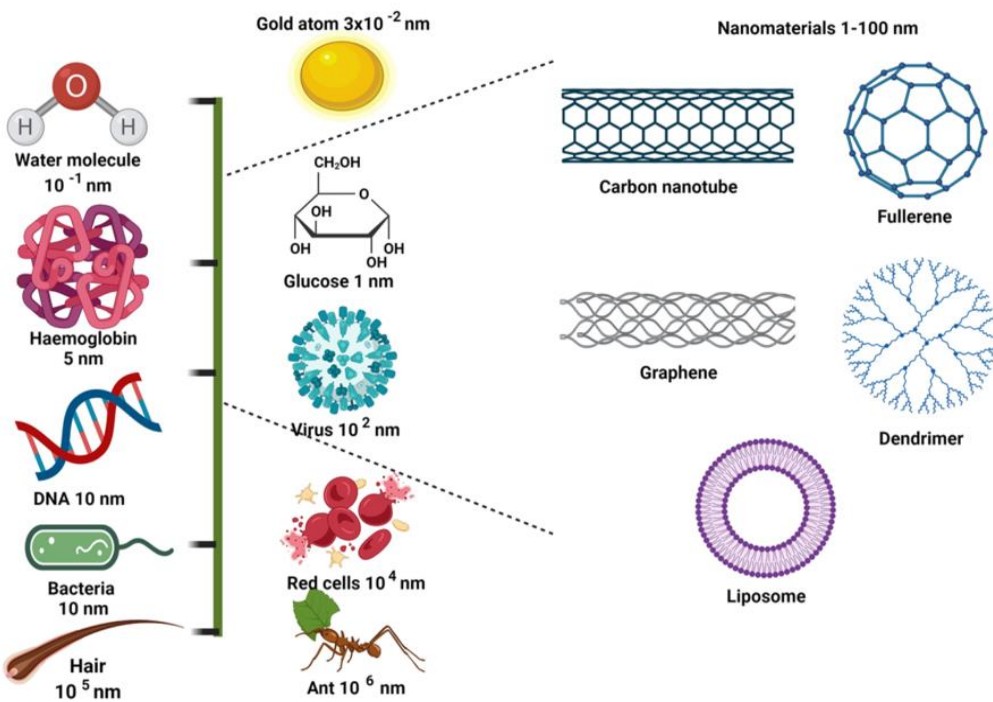

**Figure 1.** Different types of nanomaterials in nanometer range.

One of main causes of impairment worldwide is bone loss brought on by non-healing fractures following trauma, diseases like osteoporosis and tumors, which impact millions of individuals' worldwide [2]. Despite their long history and widespread use, the materials currently used in clinics include metals, ceramics and polymers (synthetic polymers) and their composites. These materials have impediments like poor biocompatibility, inadequate growth of bone and a mismatch of mechanical properties with the nearby indigenous bones. Given the growing need for innovative treatments for severe tissue loss, regenerative medicine has emerged as an appealing subject of study. For every tissue type inclusive of bone, this area of research has created innovative therapies that use cells, growth factors and biomaterials singly or in combination [3]. The creation of improved biomaterials that encourage and hasten bone rebuilding in defective areas has been the subject of numerous research studies, including the recent use of nanomaterials. In the manufacturing of bone tissue, the biomedical materials (biomaterials), which are the fundamental elements of scaffolds, are crucial. According to archeological discoveries, missing human bones and teeth were replaced with materials comprised of corals, shells, wood, animal or human bones and some metals like silver, gold and amalgam. Biomaterials are utilized in the assessment, therapy, repair, augmentation or replacement of bodily tissues or organs [4]. The scaffolds have been created using a variety of natural and synthetic polymers, including calcium carbonate, calcium phosphate and glasses. Due to the diverse scaffold requirements, composite materials made up of a combination of two or more outstanding materials are frequently employed in bone tissue engineering. Biomaterials have been widely used in the foundation of bone tissue engineering (BTE) scaffolds due to the integration, fusion and progress of the realms of medicine, biology, materials and other disciplines [5].

Numerous metabolic bone illnesses, such as age-related parietal bone atrophy, hyperparathyroidism localized infections, vitamin D-resistant rickets (VDRR) and Paget's disease can also cause bone deterioration. The persistence of so many cases related to fractures, accidents and health-related problems of bones in females, i.e., osteoporosis, has influenced various scientists and researchers to work in the domain of nanomaterials for the preparation of scaffolds [6]. The stipulation for various biomedical implants and scaffolds is becoming a huge pecuniary burden on the healthcare system due to the population's rising average age. Despite the existence of innovative techniques, traditional tissue replacements

(such as autografts and allografts) have numerous issues that must be resolved for each patient [7].

Nanotechnology has made significant strides in recent years, with the use of nanomaterials in regenerative medicine as well as in bone tissue engineering. Osteoprogenitor cell migration and recruitment is the first step in the complicated and dynamic process of bone tissue engineering, followed by the cell's proliferation, differentiation, matrix formation and bone modeling [8]. The structure and organization of the extracellular matrix (ECM) and cells that are hierarchal and span multiple orders of magnitude (nm to cm) greatly influence the characteristics of bone tissue. Therefore, novel approaches are taking into account the hierarchal assembly of tissue, from the nanoscale to macroscale, and are needed for the healing and restoration of bone defects [9]. Nanocomposites and nanomaterials are a potential platform for bone tissue engineering because they can resemble the structure of the natural extracellular matrix and ensure robust bone tissues by utilizing bone-shaped architecture. Bone itself is a class of biological nanocomposite comprised of organic constituents like collagen type I and inorganic constituents like hydroxyapatite, with a hierarchy ranging from the microscale to the nanoscale [3]. Nanocomposite scaffolds deal with the implementation of natural biopolymers that mimic the environment of the natural extracellular matrix (ECM). In bone tissue engineering, scaffolds are the substitute for the extracellular matrix. Scaffolds are composed of biodegradable biopolymers, which are porous in nature and can provide a bioactive environment in which the cells can adhere and proliferate. It also contains cells, some growth factors and certain biochemical signals, which can replace and repair the damaged tissues by creating a new environment that will allow the cells to build up their own native extracellular matrix [10]. A morphogenetic signal, receptive host cells, an appropriate carrier of the signals that can send them to specific places and subsequently function as a scaffold for the emergence of the responsive host cells, and a healthy well-vascularized host bed are all required for the regeneration of bone. This also provides structural integrity and a supporting matrix for bone regeneration. Nano-scaffolds are a 3D organization composed of polymers (natural as well synthetic) that are in the nanometer range (9–10 nm) [11]. Scaffolding materials for bone tissue engineering can be stiff or injectable, with the latter requiring surgical implantation [12]. The optimum bone tissue scaffolds for bone tissue engineering must be osteoinductive, osteoconductive and osteogenic. The requirement of osteoconductivity of these scaffolds is to promote the attachment, survival and migration of osteogenic cells [10]. Directing stem cells towards the osteoblastic lineage, osteoinductive scaffolds not only provide the physical strength but also assist in incorporating pharmacological factors [10]. To establish functional bone tissues, bone tissue engineering employs a biomimetic technique that incorporates appropriate scaffolds with pharmacological and physical stimuli, vascularization and recapturing of the hierarchical architecture of the natural extracellular matrix [13]. These biomimetic actions include selecting biomaterials found in native bone (such as hydroxyapatite and collagen), manufacturing multi-scale architectures in scaffolds, especially with nanoscale materials and incorporating growth factors such as bone morphogenetic proteins (BMPs), vascularization and/or stem cells to provide a biomimetic niche for promoting bone repair and regeneration [14].

As previously stated, the physical and chemical properties of scaffolds serve as noteworthy characteristics in their construction; thus, for the designing of scaffolds, the selection of biopolymers plays a crucial role. For the manufacturing of these scaffolds, natural as well as synthetic polymers are extensively being applied due to the variety of characteristics and bioactivities. Synthetic polymers include polycaprolactone (PCL), poly (ester amide) (PEA), polyhydroxybutarate (PHB), polylactic acid (PLA), poly glycolic acid (PGA), poly citric acid (PCA) and poly glycerol sebacate (PGS) [15]. A decellularized matrix implant can create a more pro-healing microenvironment characterized by a TH2/IL-4 immune profile, as opposed to synthetic materials like PCL that can elicit an unfavorable foreign body response owing to the release of an immune profile of T helper 17 (TH17)/IL-17 (interlukein-17) and subsequent fibrosis [16]. Additionally, they have lesser cell affinities, are biologically

inert and occasionally release cytotoxic byproducts. To mitigate these drawbacks in the preparation of an appropriate microenvironment that promotes tissue regeneration and biofunctionality, it is common practice to combine or alter synthetic polymers with natural polymers having compounds that are bioactive or have functional domains [17].

Nanomaterials are significant for encouraging cell migration and proliferation, because cells interact with tissues that are nanoscale in dimension and create a nanostructured matrix. This is particularly crucial for bone tissue regeneration in the realm of nanotechnology [18]. Bones are responsible for protection, load bearing and mobility. The framework of the body supports many organ systems throughout its lifetime. The bone has the capacity to continuously remodel and repair itself [19]. Unfortunately, defects arise from the natural repair and reconstruction of hard bone tissue; external conventions are still required to treat bone damage brought on by congenital diseases, trauma, infection or tumor resection. Therefore, a substance that is biocompatible, osteoinductive, osteoconductive and accepted by the patient's immune system is necessary for bone regeneration [20]. Bone is a hard connective tissue and reservoir of calcium and phosphate that supports internal organs in the body. Osteoblasts, osteocytes and osteoclasts are the major types of bone cells. Osteogenesis, osteoconduction and osteoinduction are the biological principles of bone regeneration [21]. The aim of novel materials utilized in bone tissue engineering is to optimize these processes. Unspecialized connective tissue cells can migrate and then proliferate into a cell lineage that forms bones as a part of the induction process. Osteogenesis is the process by which Haversian canal systems and osteoblastic cells of the transplanted bone produce new bone [22]. Vita cells are transferred directly to the regions, where new bone will grow. Osteoconduction focuses on bone formation and suggests enlisting young cells by stimulating them to become preosteoblasts. Bone development on a surface is referred to as osteoconduction, a process that occurs frequently with bone implants [14]. This review article imparts knowledge about the application of natural biopolymers and their blends for the creation of nano-scaffolds in the field of bone tissue engineering.

*Types of Biomaterials Used in Bone Tissue Engineering*

Natural polymers are the most biocompatible, biodegradable and non-toxic among all the polymers. Biomaterials derived from plants can be utilized to construct or regenerate a variety of scaffolds. Plant-based scaffolds have the advantage of being simple to assemble and manipulate, renewable, easy to mass produce and inexpensive and should therefore be explored in animal models [23]. Such scaffolds also show potential tissue compatibility and on decomposition scaffolding components do not affect the adjacent tissues. There are various forms of nanocomposite that can be prepared by the combination of different biomolecules, certain growth factors and cells. They have been employed in many applications, like resorption, degradation, vascularization and preventing infection and can also help in bone regeneration along with integration, as shown in Figure 2.

Additionally, biomaterials can be categorized based on their duration, whether they are provisional or definitive, and their mechanical characteristics, which determine whether they function as soft tissue or hard tissue. Nowadays, biomaterials can be categorized into two groups: synthetic and natural. This classification is based on their origin and is the most well-known and often-used. The latter can be further divided into metallic, ceramic, polymeric and composite materials based on the type of materials used [24]. Selecting appropriate raw materials is essential for optimum cell growth, since these may be used to prepare scaffolds with a superlative internal structure, which in turn has a favorable impact on cell activity. Scaffolds may be either synthetic or natural, but it is crucial that they maintain their characteristics throughout their whole life [25]. This type of material makes it feasible to process scaffolds with adequate control over the architectural parameters, which are crucial for migration, mass transport, cell seeding, growth and tissue formation. These parameters include the shape and size of the pore; porosity is interconnected, along with surface area and wall morphology [26]. Three factors—characteristics, qualities and behavior, i.e., reaction to environmental changes—define each biomaterial. Biocompatibility

is a crucial factor for the selection of each biomaterial for the synthesis of scaffolds. Thus, biomaterials are effective tools that do not disrupt the equilibrium state of the living thing [27].

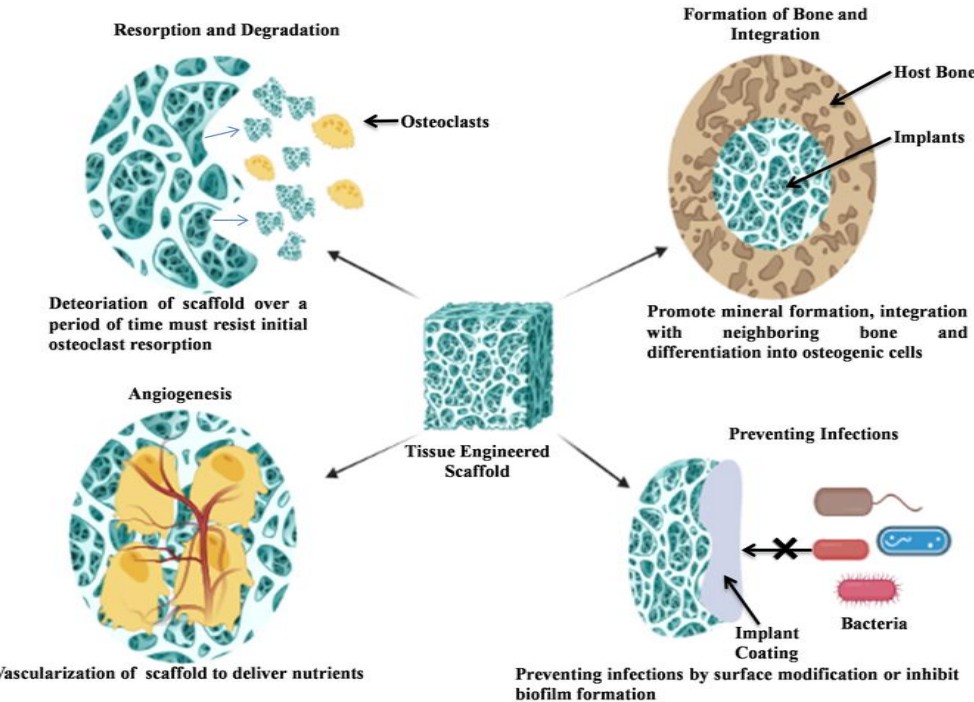

**Figure 2.** Applications of different scaffolds in the field of tissue engineering. Modified from Journal of Polymer Research [27].

## 2. Natural Polymers

Natural polymer-based nanomaterials are used for bone tissue engineering scaffolding materials due to their biodegradation and ability to stimulate regeneration of bone and tissues at both the starting and end of the process [28].

There are diverse kinds of polymeric scaffolds employed for bone tissue engineering. The natural polymers are proteins (such as soy, fibrin, silk, collagen, gelatin, actin, keratin, etc.) [29], polysaccharides (such as cellulose, starch, amylose, alginate, dextran, chitin/chitosan, hyaluronic acid, etc.) and polynucleotides (i.e., DNA and RNA). They typically exhibit enhanced biocompatibility, great cell adherence and growth promotion because of their resemblance to the elements in the extracellular matrix [25]. Hyaluronic acid, elastin, alginate, collagen/gelatin, silk fibroin, chitosan, GAGs (glycosaminoglycans), peptides and others are the natural polymers of origin for bone tissue engineering and are most frequently researched [30].

In this review paper, we especially emphasize natural biopolymers because of their cost-effectiveness, availability, negligible toxicity, very low immune response and biocompatibility. About 85–90% of the total protein in a bone is a buildup of collagen type I, which makes up the majority of the extracellular matrix [31]. Another tiny percentage of the bone extracellular matrix is composed of non-collagenous protein and polysaccharides such as osteonectin, osteocalcin, osteopontin, fibronectin and sialoprotein. About 65% of the bone is made up of inorganic minerals like hydroxyapatite (HA) and carbonated apatite (CHA). The primary cell types in bone include osteoprogenitors, osteoblasts, osteoclasts and osteocytes, and each type of cell has certain objectives [32]. Continuous remodeling of bone tissue preserves bone integrity and ensures that, over time, the bone structure and function will be preserved. Minor bone defects less than 8 mm can self-heal by bone remodeling. Furthermore, significant traumatic injuries and abnormalities brought on by tumors,

congenital illnesses and infectious diseases cannot be repaired by bone remodeling [33]. Surgical intervention and bone substitutes are required and are very important for healing large defects in bone. Autologous or allograft tissue is utilized in clinics as a substitute for bone. The main benefits of using autologous grafts are no rejection and great effectiveness. These grafts consist of the fibula, tibia and ilium [34]. The negative side effects and restricted sources of autologous bone grafts restrict their utilization. Additionally, because of the risk of rejection and disease transmission, allografts are less employed in clinics. Thus, there is a requirement of a further alternative method of bone substitution. To fulfill this requirement, the engineered-tissue bone grafts can be a better solution. The goal of tissue engineering is to create new organs and repair damaged ones by employing natural polymers. Bone tissue engineering is employed in this process to establish a framework for bone repair and remodeling using polymers and stem cells. In engineered tissue, a microenvironment similar to that existing in vivo is created by using polymer stem cells and physiochemical or biological stimuli [35]. Using natural and synthetic polymers, a biocompatible and biodegradable polymeric scaffold can be created to achieve this goal.

### 2.1. Chitosan and Chitin

Chitin is the main constituent of the exoskeleton of crustaceans and the major provenance of the natural polysaccharide known as chitosan. Chitosan is synthesized by partial deacetylation of chitin by biological, chemical or combined approaches [36]. Chitosan is not precisely defined, but generally, the degree of deacetylation of chitin of 70% or more is regarded as chitosan. By pro viding nitrogen atmosphere or by adding sodium borohydride to the solution of NaOH to prevent any form of unfavorable reactions, the process of deacetylation of chitin is performed. Chitosan has an approximate molecular weight of $1.2 \times 10^5$ g/mol. Chitosan is tough, because it has an H-bond and can be easily modified into films of good mechanical strength. Chitosan is different from chitin due to the presence of amino groups in chitosan, which provide several unique qualities [37]. Chitin has very few applications owing to its weak solubility. The chitosan D-glucosamine amino group can be protonated and soluble in a weak acidic aqueous solution with pH < 6. Chitosan is also employed in the creation of multilayered films via layer-by-layer deposition as a polyelectrolyte. In mineral acid like hydrochloric acid, chitosan dissolves in sulfuric acid, and it forms insoluble chitosan sulfate [38].

Since the only naturally occurring polysaccharide is chitosan is positively charged, it can form complexes with synthetic polymers that are negatively charged, such as poly acrylic acid; it can also form films on negatively charged surfaces, such as those found on proteins, fats, macromolecules and cholesterol [39]. In chitosan chains, hydroxyl and amino functional groups allow them to create strong covalent connections with other functional groups. Some general reactions, including etherification and esterification, can take place in the hydroxyl groups [40]. In molecular structure, both hydrophobic acetyl groups and hydrophilic amino groups are present; chitosan shows amphiphilic properties that affect the physical properties in solid and liquid states. In addition to its capacity to bind fat, chitosan also has antibacterial, antifungal, hemostatic, analgesic, wound-healing and mucoadhesive properties in both humans and animals. Chitosan can be biodegraded into a substance that is non-toxic and shows biocompatibility to some extent [41]. Because of this, chitosan can be employed in an array of medical procedures, such as topical ocular treatment and implantation. Chitosan is an excellent candidate for tissue engineering application due to all these distinctive qualities [42].

### 2.1.1. Applications of Chitin/Chitosan in Bone Tissue Engineering

Due to its versatility in being employed as fibers, sponges, and hydrogels, chitosan is employed for tissue engineering purposes. Its implementation in various fields, like drug delivery, gene therapy and as an antimicrobial agent, is shown in Figure 3. Chitosan is similar to the glycosaminoglycans (GAGs) present in the ECM (extracellular matrix) and

has significant additional characteristics that make it a perfect natural polymer to use in bone tissue engineering as a scaffold material [43].

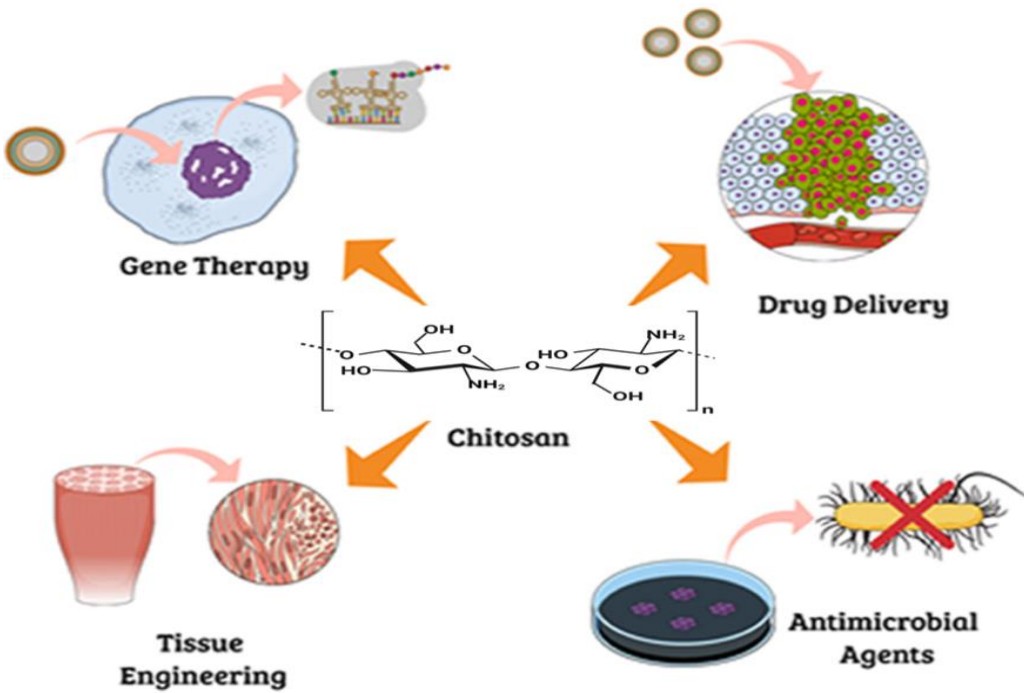

**Figure 3.** Applications of chitosan in various fields.

The significant features of glycosaminoglycans (GAGs) include various electrostatic interactions with cytokines, receptors, and cell adhesion molecules. Chitosan interacts with negatively charged GAGs and promotes cartilage chondrogenesis. Chitosan has the ability to stimulate the synthesis of special cartilage GAGs, so chitosan-based composite scaffolds are popular for the restoration of articular cartilage. Biopolymers like collagen, silk fibroin and alginate in combination with chitosan have generated interest in cartilage tissue engineering due to the non-toxicity and biocompatibility of biopolymers. Gelatin has the potential to uptake high water content, which is useful to impart alimentation to scaffolds like hydrogels and also an important element of the cartilage extracellular matrix [38].

In bone tissue engineering, materials based on carbons like nanotubes of carbon, graphene oxide and graphene have been successfully combined with chitosan. Due to their excellent mechanical qualities, these materials can be employed to reinforce organic and inorganic hybrid scaffolds. Due to the existence of free electrons, they also have antimicrobial effects and stimulate cell adhesion, proliferation and differentiation. Graphene oxide is biocompatible because of the acquired oxygen functionalities, and it can be prepared by the oxidation of graphite. Due to the presence of (OH) hydroxyl and (COOH) carboxyl groups, it is hydrophilic [44]. Aidun et al. presented a ternary composite scaffold made of chitosan, polycaprolactone (PCL) and collagen that was electrospun. As the ratio of graphene oxide was increased, a slight decrease in nanofiber diameter was observed in this scaffold. In accordance with an increase in graphene oxide, the hydrophilicity and bioactivity of the scaffold as well as cell adhesion and proliferation increased. Despite substantial research on chitosan-based bone tissue engineering scaffolds, the search for the ideal material is still ongoing, and materials are examined daily [45].

### 2.1.2. Application of Chitin/Chitosan in Bone Implants and Healing of Osteoarthritis Defects

A chitosan-poly(3-hydroxybutyrate) scaffold using β-tricalcium phosphate (β-TCP) as a reinforcement material was prepared through a novel and intriguing electrospinning

technique [46]. The electrospun scaffold had higher mechanical protection, a large surface area and high porosity. Additionally, it had crucial characteristics like biocompatibility and a rate of deterioration comparable to situations of osteoarthritis defect healing [47]. With 82% porosity, this nanocomposite scaffold demonstrated a strong tensile strength of 9 MPa. The addition of β-TCP rendered the scaffold hydrophilic, which worked as a stimulant in the growth and proliferation of chondrocytes. As compared to the normal bone, the mechanical properties of the chitosan scaffold are inferior [48]; they are incapable of supporting the load-bearing demands of bone implants. As chitosan lacks osteoconductive qualities, these scaffolds are unable to mimic the characteristics of real bones. To improve the mechanical strength and structural integrity of chitosan biocomposites for bone tissue engineering applications, biopolymers including silk fibers, alginate, chitin, polycaprolactone and polylactic acid as well as bioactive nanoceramics such as $Hap/TiO_2$, $ZrO_2$ and $SiO_2$ etc. have been created [46].

Numerous studies have found that Hydroxyapatite $(Ca_{10}(PO_4)_6(OH)_2)$ can boost the mechanical strength and osteoconductivity of implants. It is one of the more stable varieties of calcium phosphate and makes up 60–65 percent of bone. HAp also interacts with the living system and promotes the production of new bone without resorption [48]. Additionally, nanostructured HAp has greater bioactivity and a greater surface area. As a result, the chitosan and HAp composite may replicate both the inorganic and organic components of the real bone and is currently the subject of additional research [49]. Collagen, polyethylene glycol, chitosan and HAp were combined and freeze=dried using a de-hydrothermal cross-linking process to create a porous 3D scaffold. This research investigated how HAp affected chitosan-based materials. Hyaluronic acid, collagen, silk fibroin, gelatin and alginate are the other popular natural polymers that have been extensively used in this field. Moreira et al. created an intriguing in situ forming hydrogel by using gelatin, chitosan and bioactive glass. This injectable hydrogel undergoes in situ gelation in response to body temperature stimulation. This hydrogel is a novel, thermo-sensitive, minimally invasive device that may be delivered as fluid using a syringe and needle. The hydrogel's ability to be injected has added the benefit of filling small irregular holes and transporting cells and medicinal substances [50].

### 2.1.3. Application of Chitosan/Chitin in Periodontal Regeneration

The periodontium, which consists of the gingiva, alveolar bone, periodontal ligament and cement, is severely damaged by periodontitis. It is a chronic inflammatory condition brought on by bacterial infection. This illness, which accounts for more than 50 percent of cases of tooth loss in the U.S. population, is widespread around the world [51]. The formations of the periodontal pocket, and in more severe instances, infrabony (below the crest of bone) abnormalities are its defining features. Such lesions constitute a challenge to doctors in terms of treatment [52]. To combat inflammation and infection and to encourage tissue regeneration, it has been proposed to administer local delivery of active medications or substances [53]. The chitosan-based delivery method has drawn particular attention over the past ten years [54]. Many chitosan-based technologies, including micro/nanoparticles, fibers, membranes and gels, have been developed and tested under certain conditions [55]. Some interesting physical characteristics of chitosan gels may be influenced by chitosan concentration. In fact, it has been demonstrated that these chitosan-based gels (1–4%) have an intriguing viscosity that allows them to be injected within periodontal pockets. Most significantly, they can be trusted to deliver active medications to the location of the aliment. While a continuous release may be beneficial, it is crucial to remember that the kinetics of release is a characteristic behavior that is also determined by the proportion of chitosan [56]. It has also been shown that chitosan might potentially enhance the antibacterial effects of chlorhexidine, demonstrating chitosan's inherent qualities [57].

## 2.2. Cellulose

Cellulose is a versatile material with adjustable properties and can be used in systems with a wide range of biochemical and biophysical conditions. In comparison to traditional synthetic materials, cellulose-based biomaterials have several significant benefits, and they hold considerable promise for expanding the frontiers of scientific understanding [58]. Cellulose has tunable chemical, physical and mechanical properties, which is why it is an ideal candidate for biomaterial manufacturing [59].

One of the most significant and prevalent biopolymers in nature is cellulose. Cellulose is a linear syndiotactic homopolymer of D-anhydroglucopyranose that is linked by β-(1-4) glycosidic bonds. In general, cellulose is fibrillar and semi-crystalline [60]. Rapid advancement in nanotechnology offers more opportunities for cellulose. Because of features like biodegradability, renewability, sustainability, chemical stability and economy, cellulose has become popular in the form of cellulose nanofibers (CNFs), which may be used as advanced and novel biomaterials. Cellulose nanofibers are cost-effective, with generated agricultural residues or wastes providing nanosized reinforcements in composite materials [61]. The cellulose nanofibers that are segregated from lingo-cellulosic plants typically show distinctiveness primarily through a high modulus and high surface area. They exhibit enormous potential for use as reinforcement in biopolymer environments. Due to their similarity to numerous biomaterials, polymeric materials and metal oxides, cellulose nanofibers have a high crystalline content, making them ideal fillers. The leading edge of nanotechnology, material science and biological science is represented by the creative production of hybrid nanostructured materials known as cellulose nanomaterials. The most common method to prepare cellulose nanocrystals is acid hydrolysis [62]. Cellulose synthase proteins, which catalyze glucan chain polymerization, are present in all cellulose-synthesizing species, including bacteria, algae, tunicates and higher plants. The vast differences in the lifestyle of the organism and the structure of the cellulose it produces suggests that regulatory proteins and the underlying mechanisms of cellulose synthesis may have evolved independently, even though the catalytic domains of cellulose synthase are preserved in all cellulose-synthesizing organisms [63]. Cellulose obtained from plants can be converted into cellulose microfibrils. Plant cellulose and bacterial cellulose are the two main classes of cellulose, and both have good applications in tissue engineering, as depicted in Figure 4. Bacteria produce bacterial cellulose, which is a natural biomaterial. It has outstanding mechanical properties, water-holding capacity and suspension ability. Additionally, it possesses great biocompatibility and biodegradability as well as a high degree of crystallinity and high purity. As a result, bacterial cellulose has gained considerable interest from both academia and industry. Plant cellulose frequently contains impurities, including lignin, pectin, hemicellulose and other substances, whereas bacterial cellulose is nearly pure, has much higher water content and exhibits noticeably increased tensile strength because of its longer chain length [64].

A mixture of polycaprolactone (PCL), gelatin (GEL) and bacterial cellulose (BC) reinforced with various quantities of hydroxyapatite (HA) nanoparticles was used to create a unique porous scaffold for 3D printing [65]. Four distinct composites were created, with infill rates ranging from fifty to eighty percent, to obtain ideal pore size and a homogenous blending ratio. At the 80% infill rate, the optimal pore size for a bone tissue imitating ECM was attained, producing a uniformity ratio of more than 90%. In comparison to composites devoid of bacterial cellulose, bacterial cellulose-containing composites displayed a lower tensile strength and a higher cell survival rate.

Additionally, compared to the other composites, adding 0.25% HA to the blends improved cell adherence and vitality. Comparative research on these printed scaffolds showed that they might be used as bone implants [66].

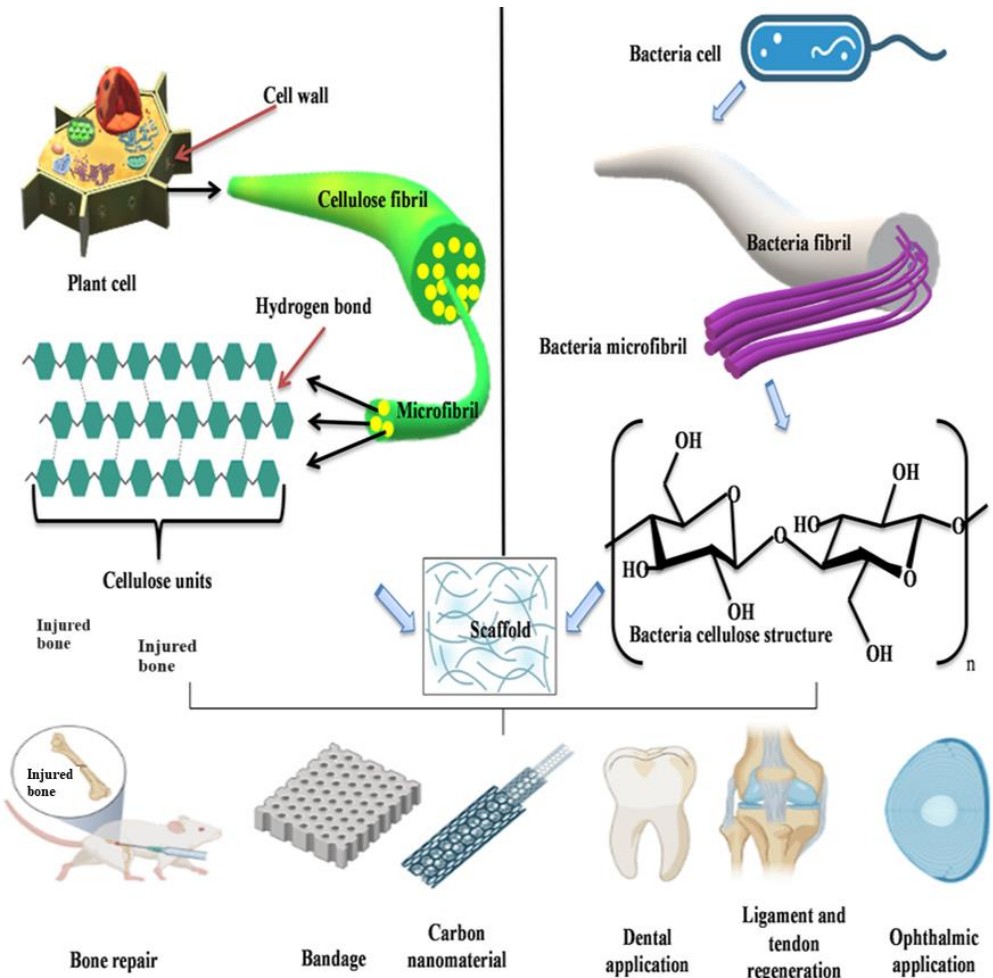

**Figure 4.** Sources of plant and bacterial cellulose and its application in tissue engineering.

### 2.2.1. Application of Plant Cellulose in Bone Tissue Engineering

Cellulose derivatives are now being used in biomedical applications. In the field of biomaterials, the use of cellulose for membranes in biomedical application is very important. For the preparation of membranes for osseointegration, cellulose acetate is proven to be an appropriate biomaterial due to its excellent biocompatibility [67]. When cellulose acetate breaks down, it produces fragments that are not cytotoxic at the site of implantation and an acetyl group that slightly alters the local pH. These cellulose acetate membranes were employed to cover magnesium alloy implants, and they produced positive results for both in vitro tests and after being implanted into the mice's intermedullary canal [68]. In the osseointegration process, cellulose acetate (CA) prepared with sericin or resveratrol was successfully applied [69]. In MC3T3 osteoblast precursor cells, the proliferation ability implicitly improved the osseointegration of implants covered with cellulose acetate membranes. To avoid any post-operative infections, especially in dental areas, controlled antibiotics were applied at the implantation site. For such uses, dual role membranes that promote osseointegration and permit the release of the drug doxycycline were developed [70]. For hemodialysis, composites of cellulose acetate (CA) were created using fillers like silica nanowires or hydroxyapatite or by combining carbon nanotubes and graphene. All of them had a non-cytotoxic character, high bovine serum (BSA) binding ability and high flow rates. Cellulose acetate has some other applications, like in the preparation of membranes having antibacterial properties or antibiotics from aqueous solutions [71].

Salehi et al. investigated decellularized cabbage (DCB) as a natural cellulose material for bone tissue engineering. By using Triton X100, SDS (sodium dodecyl sulphate) and sodium hypochlorite, they decellularized cabbage and then washed it with deionized water, PBS (phosphate buffer saline) and hexane. The shape of the initially produced DCB scaffold was asymmetrical, and the planted cells were entrapped. The tensile strength showed that decellularized (DCB) scaffold had the ability to resemble the spongy bone structure. According to in vitro studies, the bone marrow-derived mesenchymal stem cells (MSCs) grown on the scaffold showed remarkably higher levels of alkaline phosphatase (ALP) activity and mineralization than bone marrow-derived mesenchymal stem cells cultured on tissue culture plates, which served as control group. Gene expression cells grown on decellularized cabbage leaves exhibited higher levels of osteocalcin, collagen-1, Runx-2 and ALP expression than cells cultured in Petri dishes. Based on the above discussion, it can be concluded that decellularized cabbage is a remarkable ECM-mimicking structure for bone tissue engineering application, considering the osteogenic capacity and bone healing ability of the scaffold [72].

### 2.2.2. Applications of Bacterial Cellulose for Bone Regeneration

A wide range of research has been conducted using cellulose from bacteria (BC) in guided bone regeneration on several animals. Noncritical bone defects were evaluated in vivo in rat tibiae on a bacterial cellulose membrane, and these defects were entirely filled by fresh bone tissue after four weeks [73]. Studies on the impact of BC membranes on bone regeneration on rat skulls produced results that were equivalent, showing that after 8 weeks of implantation, new bone had formed on both the periphery and the core of the bone defect [74]. The study of bacterial cellulose membranes as a barrier membrane for directed bone regeneration on rat calvarial defects did not cause an inflammatory reaction and preserved appropriate spaces for bone regeneration [75], and no signs of a foreign body reaction were noticed when bacterial cellulose grafts were utilized to treat rabbit nasal dorsum, which fragmented in six months, a favorable sign of bacterial cellulose integration [76]. At day 14 after implant, mineralized formation of bones was seen on the exterior and inner surface of the femoral cortical bone of dogs [77].

To treat maxillary canine periodontal abnormalities in beagle dogs, a bacterial cellulose composite membrane was utilized as a guided tissue regeneration membrane. The results reveal that the membrane encouraged periodontal tissue regeneration by generating new bone [78].

### *2.3. Albumin*

One of the most important and fundamental study fields in nanomedicine is the use of peptides and proteins. Experts from different fields, like nanobiotechnology, pharmacy, toxicology, nanomedicine, immunology and other medical sciences, are currently examining the many facets of these essential biomolecules, including understanding how the resulting nanostructures interact with the body, and are using them for therapeutic and diagnostic purposes. The animal protein albumin has several therapeutic applications [79]. Because albumin contains several functional groups, it can interact with a large number of drugs. The ability to fully recognize albumin's structure and amino acid sequence as well as its multiple charged groups enables the binding of different drugs to albumin nanoparticles by a variety of mechanisms, including electrostatic attraction with negatively charged drugs and positively charged drugs like oligonucleotides and dual compounds. The albumin-based nanoparticulate structure is used in the administration of medicines in the categories of anticancer agents, vaccines, hormones, anti-asthmatics and anti-inflammatory agents [80]. The most prevalent human serum albumin has been reported to influence cell attachment to different scaffold materials in a way that is similar and superior to collagen and fibronectin following particular treatments. It may act as a bridge between the scaffold and the cells, facilitating the fusion of these two elements. With little variation in the sequence of amino acid content, it is present in the white portions of eggs, milk, cow blood, pig blood serum,

sheep blood serum, human blood serum, human liver and many other animal and plant tissues [81]. Bovine serum albumin (BSA), human serum albumin (HSA), porcine serum albumin (PSA) and ovalbumin are the major types of albumins most often used in tissue engineering scaffolds. The most prevalent protein in plasma is serum albumin; it has tremendous importance because it regulates a variety of vital processes, including blood pH and osmotic pressure. This globular protein albumin has several binding sites and is non-toxic, highly soluble in water and stable from pH 4 to 9 as well as in organic solvents. Albumins detoxify blood plasma, lowering the amount of dangerous chemicals such as heavy metals, reactive oxygen species (ROS) and ions. It also works as a solubilizing agent for fatty acids [82].

Yamaguchi et al. produced a significant body of research on the use of albumin in bone regeneration trials. Firstly, they demonstrated that local albumin production increases following bone fracture. Additionally, they examined the effects of supplementing a medium with bovine serum albumin on bone explants in vitro. They discovered that albumin enhances the calcium and DNA content of the fragments of bone [83]. Sources of albumin, like human serum albumin (HSA), bovine serum albumin (BSA) and ovalbumin (OVA), can be used to prepare scaffolds in in vitro as well as in vivo analysis for bone healing by blending it with different polymers and adding some growth factors, as shown in Figure 5. A variety of sources, including eggs (ovalbumin), bovine serum albumin, human serum albumin, rat serum, soy, milk and grains all are used to obtain albumin [84]. They are biocompatible and have high stability, low temperature and pH sensitivity. For biophysical and biochemical research, BSA, HSA and OVA (ovalbumin) are most frequently used, and all three albumins are commercially accessible. The presence of sulfhydryl groups and disulfide bonds in albumins permit interaction with both organic and inorganic ligands [80]. OVA, HSA and BSA have similar characteristics, and all are primarily composed of α-helix, with a globular structure. They have both hydrophilic and hydrophobic sites and exhibit an acidic nature.

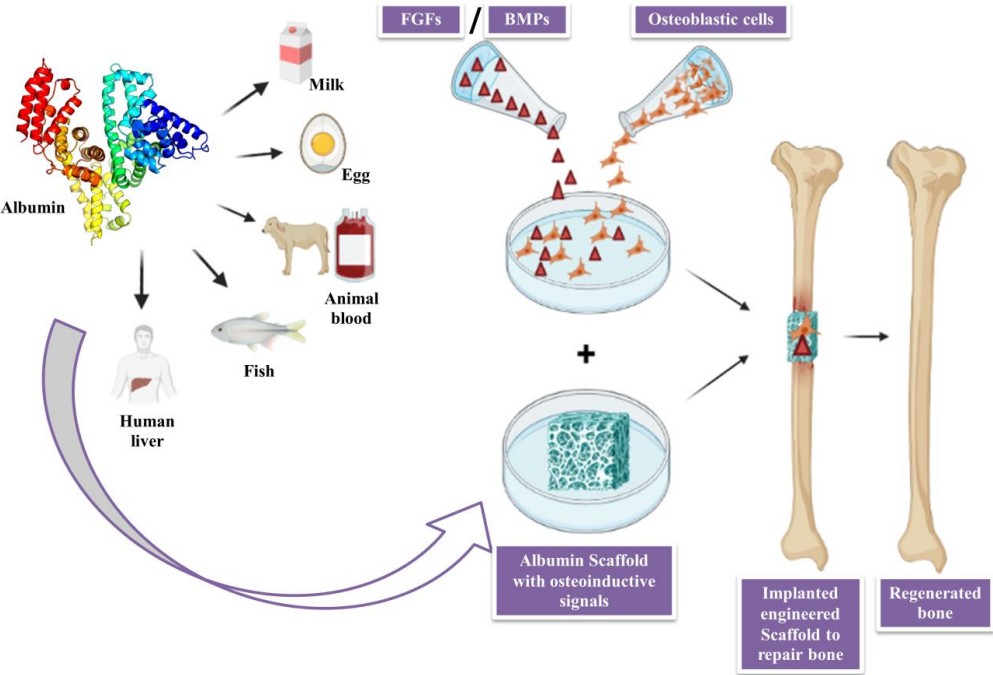

**Figure 5.** Sources and preparation of albumin scaffold for bone regeneration.

Ovalbumin (OVA) is a sequence of a single 385 amino acid polypeptide chain with a spring-like structure and a helical reactive loop configuration. Ovalbumin can form gel and foam, which makes it a common ingredient in the food industry [85]. Ovalbumin is one of the first proteins identified and accounts for 60–65% of the protein in avian white eggs.

Its structure and properties show that it is a member of the serpin superfamily of proteins. Despite sharing a 30% sequence identity with antitrypsin and other inhibitor proteins of the serpin family, ovalbumin does not exhibit any protease inhibitory function [86].

Bovine serum albumin (BSA) is a protein made up of three homologous domains and two subdomains. It has 583 amino acid residues and has 67% α-helical structure. It is a 66 kDa globular protein and is procured from the blood of cows. BSA is a vital part of tissue media used for culture, and a scaffold based on albumin will continue to be an excellent substrate, since it provides structural support for cells and tissue engineering [87]. The three corresponding domains, I, II and III, which make up the three-dimensional structure of BSA are selective for fatty acids and metals. Each domain is the result of two sub-domains, with most of them helical. Sulfide bridges are used to connect these sub-domains widely [88]. Due to its accessibility, binding affinity in the development of ligand protein complexes, intrinsic fluorescence caused by its two tryptophan residues (Trp 134 in the first domain and Trp 212 in the second domain) and structural and functional resemblances to HSA, BSA is intriguing. Human serum albumin (HSA) has 585 amino acids in a globular conformation, the same as BSA, and three homologous domains that are shaped like an α-helical ellipsoid [89]. The amino acids in HSA are preferentially absorbed by tumors and inflammatory tissue, and they can be dissimilated and utilized as a source of sustenance for peripheral tissues [90]. Additionally, HSA has a variety of binding sites, which enables it to transport ions, medicines, hormones and fatty acids throughout the body. These qualities make it a strong candidate to be used as a drug delivery system and in pharmaceutical formulations [91]. In terms of structure, characteristics and conformation, BSA and HSA are similar. The tryptophan residue position is the primary distinction between BSA and HSA, which share 76% sequence similarity. HSA only contains one Trp at position 214, but BSA contains two tryptophans at two different positions, at Trp-212 and Trp-134 in solvents. The differences between ovalbumin and HSA/BSA in terms of weight, size structure, S-S bond bridges and SH groups are remarkable [92]. BSA and HSA are protein carriers utilized in various delivery methods, because they are less immunogenic than other albumins and are thought to be better tolerated by humans. Currently, BSA and HSA are used in drug delivery systems in which drugs are conjugated to the binding sites of proteins, improving drug water solubility, therapeutic efficacy, bioavailability, biocompatibility and biodistribution as well as lowering adverse drug reactions, including anti-inflammatory, chemotherapeutic and hypoglycemic drugs [93]. The amino acid of HSA is made up of one polypeptide strand, which combines to create three different secondary structures. Thirty-five cysteine roots make up albumin, and they play a crucial part in this protein by generating 17 disulfide linkages. Additionally, a large number of charged amino acids, including lysine, arginine, glutamic acid and aspartic acid, are present in albumins and fabricate its structure, which is foremost for the proteins in many biological functions as well as for the formation of nanoparticles and the binding of various substances [90]. By using X-ray crystallography, the three-dimensional structure of human albumin has been identified. The albumin has a heart-shaped morphology, with dimensions of 80 by 30 angstrom. Around 80% of the plasma osmolarity pressure is generated only by albumin, which also has a significant buffering effect on blood pH. The usual half-life of this protein in human blood serum is 19 days, and it is produced in the liver, along with many other plasma proteins, at a daily rate production of 10–15 grams. Fatty acids, eicosanoids, bile acids, steroid hormones, vitamins C and D, foliate and magnesium, as well as numerous drugs, like penicillin, sulfonamides and benzodiazepine compounds, work as carriers of many molecules through albumin [94]. The protective function of albumin is to bind with toxic substances that are of external origin, like benzene, other carcinogenic compounds and various others. Human serum albumin is applied as a therapeutic agent to treat a variety of ailments, including shock, burns, albumin deficiencies, trauma, cardiac surgery, acute respiratory issues and blood dialysis [95].

2.3.1. Applications of Albumin in Bone Tissue Engineering

The cross-linked albumin scaffolds have high wettability, which promotes craniofacial regeneration. When combined with collagen I, they encourage the development of mesenchymal stem cells into osteoblasts, has a very porous structure, resilience, moderate mechanical strength and good compatibility. The mechanical strength of albumin scaffolds produced by heat aggregation has been reported to be 8.5, with pH 12 showing the highest biodegradation rate and with pH 4.8 showing the lowest biodegradation rate [96]. Another technique involved electrospinning albumin to create a scaffold, which was found to be nontoxic, biodegradable and supportive of endothelial muscle cell adhesion in vivo. The functional group on the electron fiber facilitates protein conjugation, which can promote physiological processes that aid in the growth of tissue, like the lungs. For protein in general and albumin in particular, cross-linking, freeze-drying, heat-aggregation and electrospinning techniques have been shown to be effective in manufacturing the scaffold.

The use of albumin scaffolds in bone tissue engineering has been effectively demonstrated and utilized. They have high seeding efficacy, are biocompatible, non-immunogenic and affordable, and have regulated decomposition [97]. Successful fabrication of albumin fiber scaffolds with mechanical characteristics like heart tissue has been accomplished. When compared to PCL fibers, these fibers perform better and act as scaffolds for the fabrication of functioning cardiac tissues. Cell adhesion is encouraged by this cardiac scaffold's capacity to bind serum proteins, and the functional group on the albumin scaffold facilitates protein conjugation, increasing physiological processes and promoting tissue growth [98]. However, further research needs to be conducted on serum protein binding, the release of therapeutic biomolecules that will enhance tissue function, and the capacity of the tailored patches to enhance heart functions following infraction [99].

In addition to providing structural support to connect the injury site, biodegradable polymer scaffolds also help to direct axon regrowth. In experiments designed to determine the potential and safety of the new albumin scaffold, human serum-derived spongy scaffolds could enhance the recovery of motor functions in spinal cord injury (SCI) in rats [100]. The difficulties in reproducing the normal dynamic of the integrated network of fundamental cells, function and orientation of its ECM, perfusion-ventilation relationships and immune response, which are all necessary for ideal lung function, will be in creating complex 3D functional lung tissues ex vivo. The use of albumin as a surface grafting material for periodontal intrabony defects has been the subject of various investigations, resulting in cementum resorption and regeneration [101]. Albumin-coated bone chips can mediate tissue regeneration; however, seeding bone grafts with cells or grafting them with collagen or fibronectin is insufficient for regeneration. The active functional groups present in the albumin may be the cause of this quick regeneration process. Since the surface marker on the lung will work with the albumin scaffold, isolated albumin can be grafted on the lungs before seeding with stem cells. Since they are simple to produce, biodegradable and biocompatible with many cells, scaffolding materials and non-immunogenic albumin scaffolds are an autogenic biomaterial that can be used indefinitely. The use of albumin-based biomaterials in bone, cardiac and neural tissue engineering has been well established. This biomaterial can also be grafted into the decellularized lungs prior to recellularization [95].

2.3.2. Role of Albumin as Biomaterial in Regenerative Medicine

Gold, steel, platinum, titanium and other metal-based biomaterials are ideal because of their inertness and structural capabilities; nevertheless, their surfaces lack bioactivity [102]. Despite its lack of flexibility, bio glasses and ceramics, with great biocompatibility and strong mechanical qualities, find use in dentistry and bone regeneration [103]. Some synthetic polymers like PLA, PCL and PGA are used for the preparation of scaffolds, medical devices and implants. They have good physical characteristics, moderate rates of degradation and customizable design, but they do not encourage cell adhesion and spreading [102]. Utilization of natural components, blood-derived products, decellularized organs, primary cells and stem cells is an innovative technique for improving biological compatibility. Albu-

min is a blood-derived protein substance having the potential for autologous or allogeneic tissue engineering and has been successfully used in the production of scaffolds, hydrogels and coatings [91].

Further in vivo testing of albumin coating on bone allografts involved creating bone lesions in the parietal bones of old female rats. The bone defects were filled in with bone albumin and a non-coated demineralized bone matrix, and mechanical tests, CT (computed topography) and micro-CT were performed to access the graft's integration with the surrounding tissue. The bone development was more rapid, more durable and had two times greater fracturing force compared to the uncoated bone grafts, according to in vivo computed topography and ex vivo micro-computed topography tests. The grafts with albumin coating attracted around twice as many cells as the grafts without albumin coating when MSCs were incubated in vitro [104].

### 2.3.3. Role of Albumin as a Nano-Scaffolds

When used in biomedicine, albumin has various benefits; nevertheless, for some reasons, albumin works best when coupled with other biomaterials [105]. Due to their resemblance to the inorganic components of bone tissue, bioactivity, osteoconductivity and mold ability, calcium phosphate-based biomaterials are commonly employed in biomedicine, particularly for bone regeneration [106]. The production of (ACP-PLA) ACP-poly (l,d-lactic acid) nanofibers with the addition of BSA (bovine serum albumin), reported by Fu et al., utilized amorphous calcium phosphate (ACP) nanoparticles. The ACP-PLA solution containing BSA was easily electrospun into nanofibers with a fibrous structure. The PLA, PLA-ACP and BSA containing ACP-PLA nano-fibrous round mats were submerged in a simulated body fluid with ion concentrations that were approximately identical to those in human blood plasma to promote mineralization. Except for PLA, the shape of the nanofibers had already changed after just one day of immersion due to the deposition of tiny nanoparticles on their surfaces. Increased inorganic matter deposition and the development of a nanosheet network were produced when the mineralization time was extended. The total hydrophilicity of the surfaces of the BSA containing ACP-PLA and the ACP-PLA alone was presumably brought on by the enhanced water adsorption inside the recently constructed porous nanosheet network. After a week of cultivating MG-63 cells (human osteosarcoma cell lines) on the scaffolds, a steady rise in cell metabolic activity on all the examined fibers was observed. Water-soluble drug-delivery methods for tissue engineering may be able to use the hybrid material with BSA and ACP-PLA, since it quickly mineralized and had great biocompatibility [107]. Patel et al. and Haag et al. studied the intricate relationships between BSA and calcium. When albumin is exposed to calcium, it undergoes several structural changes that increase its bioactivity and have several positive impacts on bone tissue regeneration, according to their modeling research [108].

### 2.4. Silk Fibroin

Silk fibroin is a kind of protein that is derived from silkworms. It has great biodegradibility, bioresorbability and biocompatibility, low immunogenicity and tunable mechanical properties. Due to its unique properties, silk fibroin is used as a potential biopolymer for bone tissue engineering [109]. The integration of silk fibroin with other biopolymers to create silk fibroin composite scaffolds can encourage cellular behaviors like cell differetiation, cell proliferation and cell adhesion. Additionally, silk fibroin-based biomaterials can be made into a variety of materials formats, including hydrogels, sponges, 3D structures, films and nanoparticles. It is a natural protein made up of amino acids in which 90% of the amino acids are glycine, alanine and serine, which form crystalline β-pleated sheets in silk fibers [110].

There are numerous taxonomic families that produce silk, including those of silkworms, glowworms, lacewings, mites and spiders; some of these species have the ability to spin silk into threads for cocoon regeneration [111]. According to recent research by Yoshika et al., the psychidae family, popularly called Bagworm moths, is expected to create

the hardest type of moth silk, which is used currently [112]. The most often used types of silk for biological applications are from spiders and silkworms. In spider silk, when the silk is spun and when it comes in contact with the air, it hardens, which makes it difficult to generate a large quantity of silk fibers. Compared to spiders, the yield of fibers obtained from one silkworm cocoon is around 10 times as much [113]. The two families of silkworms on either the mulberry tree (Bombycidae) or alternative food sources known as non-mulberry (Saturniidae) silks are recognised to play the most significant roles in the study of silk produced from silkworms. *Bombyx mori* is a mullbery-feeding silkworm; it produces a higher quality of silk than saturniidae. It is the most widely used silkworm, producing large amounts of silk with better quality [114]. The cocoons of *Bombyx mori* contain impurities called sericin (16.7–25%) and a protein known as silk fibroin, as shown in Figure 6 (75–83.3%). Silk fibroin is a protein with a semi-crystalline structure that is mostly used for its ability to support weight. On the other hand, sericin is an unstructured protein polymer that acts as a gumming substance [115]. Sericin-free fibroin has been discovered to have superior mechanical qualities over fibroin wrapped in sericin, which has a 50 percent increase in tear strength, a 15–17 GPa modulus and a stress break reaching up to 19%. To produce pure silk fibers of a higher quality, scientists and many researchers are working to improve the degumming procedure, which normally calls for reagents and organic solvents. The traditional Marseilles soap degumming method has been supplanted by the sodium carbonate degumming method, which is presently the most popular method due to its speed and affordability [116].

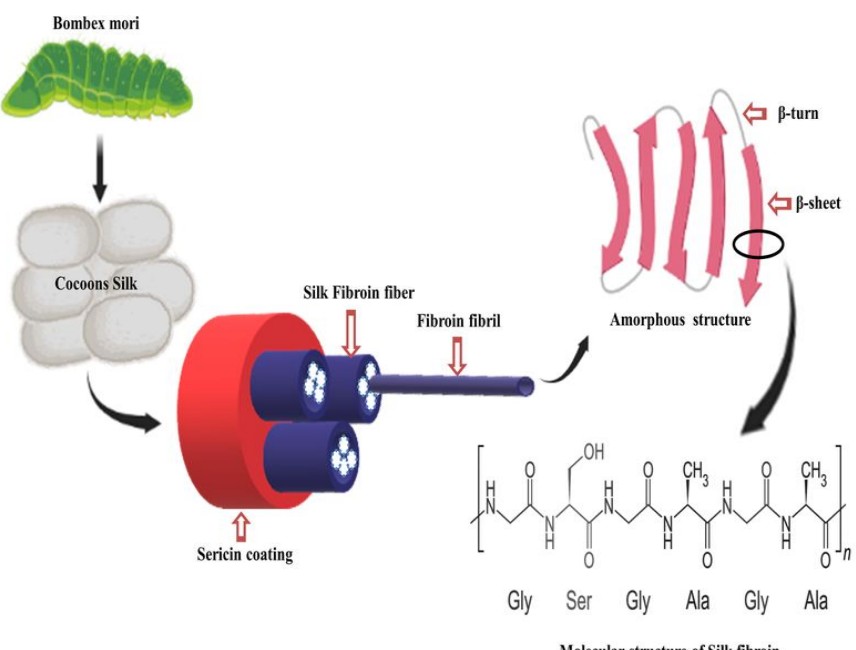

**Figure 6.** Sources and structure of silk fiber.

The structure of silk fibroin extracted from the cocoons of *Bombyx mori* is shown in Figure 6. Light (L) 26 (kDa) and Heavy (H) chain (390 kDa) are the two major chains that make up silk fibroin, and they are joined by a disulphide bond to create the H-L complex. The three polypeptides that make up the cocoons of *Bombyx mori* are the L-chain, H-chain and P25, which are each visible at a molar ratio 6:6:1. Glycine 45.9%, Serine 5.3%, Alanine 30.3%, Valine 1.8%, and 15 other amino acid types 4.5% make up the H-chain amino acid sequence [117]. The H-chain is composed of 60–75% repeats of the Gly-X dipeptide motifs. Dipeptide repeat hydrophobic residues can form stable antiparallel β-pleated sheets. The two hexapeptides whose peptide sequences are Gly-Ala-Gly-Ala-Gly-Ser and Gly-Ala-Gly-Ala-Gly-Tyr make up 70% of the GX dipeptide motif region. Crystalline and amorphous

structures can be found in the secondary structure made from solutions of regenerated silk fibroin (RSF) [118]. When silk is crystalline, it has β-turns and an insoluble structure made of folded β-sheets, whereas when it is amorphous, it contains helix turns and random coil structures [119].

### 2.4.1. Application of Silk Fibroin in Bone Tissue Engineering

A specialized connective tissue, bone comprises 60% inorganic matrix and 35% organic components. Collagen makes up more than 90% of the organic extracellular matrix of bone. The remainder is made up of hyaluronan, proteoglycans, bone sialoprotein, osteopontin, osteonectin and osteocalcin. Most of the inorganic mineral phase of bone is made up of (hydroxyapatite), with the remainder being made up of inorganic salts and carbonate. This indicates that HA and collagens are the primary building blocks of bone tissue, which improves the strength and hierarchical structure of bone [120]. For application in bone tissue engineering, scaffold materials should ensure matrix toughness and permit extracellular matrix deposition. Silk fibroin has been extensively researched in bone tissue engineering and has demonstrated biocompatibility as well as great toughness and mechanical strength. To improve osteogenic characteristics, regenerated silk fiber (RSF) scaffolds are combined with additional biomaterials, like collagen or calcium phosphate-based inorganic components [121]. The FDA has recognized the growth factors known as bone morphogenetic protein (BMP-2) and BMP-7 as supporting bone development and regeneration. Regenerated silk fibers (RSF) have been demonstrated to exhibit improved osteoblast adhesion and differentiation, increase alkaline phosphatase activity and support bone formation in vivo when combined with these growth factors and human mesenchymal stem cells (HMSCs) [119].

### 2.4.2. Silk Fibroin Utilization in the Formation of Tendons and Ligaments

Ligament and tendon tissues are formed by collagen and fibrocytes, resulting in a thick fibrous connective tissue that is quickly damaged and has a severely limited capacity for endogenous regeneration. Regenerated silk fibroin (RSF) scaffolds have developed into a popular biopolymer for the use in ligament and tendon tissue engineering because of their distinctive mechanical qualities, such as high toughness as well as good elasticity and structural stability. The first RSF matrix was successfully used in 2002 to create an anterior cruciate ligament (ACL) that had mechanical qualities like those of the human ACL [122]. In contrast to the RSF hydrogel knitted scaffold, web-like SF sponges were created by Liu et al. for knitted scaffolds in which HMSCs were seeded. The findings show that web-like microporous RSF sponges can increase cellular activity, while SF-based knitted scaffolds had established structural robustness [123]. Additionally, it has been demonstrated that immobilizing an arginine-glycine-aspartic acid (RGD) scaffold may encourage the attachment of BMSC cells, resulting in increased levels of human bone marrow cells and ligament fibroblast production. Furthermore, successive injection of growth factors such as basic fibroblast growth factor, transforming growth factor and epithelial growth factor lead to the proliferation and differentiation of BMSC cells on RGD-coupled RSF scaffolds, which could aid in the creation of ligament tissue [124]. Including cell differentiation towards a fibroblast lineage, increased matrix in-growth and collagen formation, growth factors boost biochemical and mechanical properties. In this way, it can be observed that, combining RSF with synthetic materials such as polyelectrolyte and PLGA and natural biomaterials including collagen type I, hyaluronic acid and gelatin enhance the properties of scaffolds for use in the repair of ligament and tendon connective tissue [115].

### 3. Importance of Polymers for Local Drug Delivery Systems in Oral Cavity

Biomaterials infused with drugs can be fabricated by using various methods, such as solvent casting, hot melting extrusion, rolling techniques, direct compression and 3D printing. The optimal method to employ will be contingent upon the characteristics of the polymer, the desired structure of the biomaterials, its intended usage location and the

necessary drug dosage for the treatment [125]. Additionally, the discussion will encompass gels, mucoadhesive tablets, microneedles and films. The primary categories of biomaterials utilized for drug delivery in the oral cavity predominantly include fast dissolving films and gels, microneedles, films and mucoadhesive tablets [126]. An oral fast-dissolving film is a type of drug delivery system that is designed for ease of use and rapid absorption into the body. The thin film is placed in the mouth, usually on the tongue or inside the cheek, and it quickly absorbs saliva, hydrating and adhering to the oral mucosa. This allows the film to dissolve and disintegrate, releasing the medication directly into the blood stream via the rich network of blood vessels in the mouth. This method of drug delivery is often preferred for its convenience, rapid onset of action and the ability to deliver a precise dose of medication [127].

Mucoadhesive films are one of the formulations that are frequently used for transbuccal drug delivery. The huge surface area and high flexibility suggest that there is a greater area for drug absorption. Other avenues besides mucoadhesive films include pastes, sprays, gels and pills. Because they are more likely to be absorbed by saliva than gels and other formulations like pastes and sprays, they also ensure a higher dosage of the medication [128].

Microneedles are the 3D microstructure that may penetrate sufficiently deeply into tissue to administer drugs without disrupting the surrounding tissue's nerves or rupturing blood vessels. Therefore, the use of microneedles in treatment allows for the minimally invasive transport of many molecules to the tissues, thereby circumventing the drawbacks of traditional transdermal drug administration techniques [129]. These biomaterials also have the benefit of being versatile, since they can be utilized to deliver medications to the eyes, skin and oral cavity [130].

## 4. Utilization of Biomaterials for Bone Loss and Regenerative Medicines in the Oral Cavity

According to Pouroutzidou et al., the present advanced research is primarily focused on the engineering of electrospun membranes utilizing Poly(lactic-co-glycolic acid) (PLGA). These fabricated membranes are explicitly intended for a sophisticated process known as periodontal regeneration. This process is crucial in the restoration of tissues existing in the periodontium. The novelty of this cutting-edge research lies within the integration of moxifloxacin, an antibiotic encapsulated within silica-based mesoporous nanocarriers. These mesoporous silica-based materials are widely recognized, due to their inherent property of controlled release. In this specific context, these materials are anticipated to function as efficient carriers for the controlled delivery of moxifloxacin. The potential implications of this study are expansive, potentially leading to the development of innovative drug delivery systems specifically designed for periodontal applications. This research strives to advance the field of periodontal regeneration, opening new avenues for future explorations.

This membrane derived from Poly(lactic-co-glycolic acid) (PLGA) through the electrospinning process exhibited compatibility with erythrocytes when their mass was below 1 mg. The research noted that an escalation in the quantity of the polymer led to the development of more consistent fibers, characterized by larger diameters and pores.

Moreover, the examination of electrospinning parameters by researchers unveiled that an elevation in both applied voltage and collector rotation speed resulted in more uniform fibers, featuring increased diameters and larger pores. The deliberate adjustment of electrospinning variables appears to exert control over the physical characteristics of the membranes, rendering them more conductive to application in periodontal regeneration [131].

## 5. Importance of Inorganic Biomaterials in Combination with Polymers

Hard tissue engineering has seen bioceramic scaffolds garnering much consideration because of their remarkable capacity to restore and rebuild diseased or damaged tissues. They are stable, biocompatible and have high strength at compression. Nevertheless,

fragility, low resilience, low tensile strength, production complications and sluggish decomposition are a few of the drawbacks of bioceramics. By employing composite materials, these restrictions can be removed. Bioinert, bioactive and bioresorbable are the three main categories of bioceramics. The best bioceramic materials for repairing and rebuilding bone tissues are hydroxyapatite, alumina, calcium phosphate, zirconia, tricalcium phosphate and bioglass carbon. Because of its exceptional bioactive and bioresorbable qualities, hydroxyapatite is well-utilized in orthopedic bone repair. It can be synthesized in a variety of forms, including ceramic plates, blocks, powder and granules, for usage in a variety of bone tissue applications [132].

The organic–inorganic natural structure of bone is mimicked by nano-bioceramics' polymeric structure, which offers nanosized crystalline materials the ability to biomineralize and function as nanoscale hydroxyapatite crystals embedded in a polymeric matrix, much like the collagen phase of the bone. Many inorganic biomaterials have been employed to restore tissues that are damaged; silicate biomaterials, like bioactive glass $CaSiO_3$ and Ca-Si-M (where M = metals like Mg, Zr, Ti, etc.) ceramic systems, have the unique ability to release Si ions in a concentration that stimulates osteoblast growth, making them appropriate for use in the regeneration of bones [133]. The mechanism of osteogenesis (Mg and Ca) and angiogenesis (Cu) is stimulated by certain metal ions, which are also required for the synthesis of enzymes; hence, they promote the development of bone. Because of their anti-inflammatory and antibacterial qualities, the use of various types of metal ions, like silver, copper and zinc, results in additional therapeutic benefits [134]. Covalent bonds can be established between organic and inorganic constituents, resulting in the generation of composite materials. An instance of this can be observed in the composition of bone, wherein the organic compound collagen is connected through covalent linkage to the inorganic substance hydroxyapatite [135].

In particular, within the bone structure, collagen fibers are intricately integrated into a matrix featuring hydroxyapatite crystals, a mineral composed of calcium phosphate. The amalgamation of these components establishes a harmonious equilibrium between the flexibility inherent in the organic element and the hardness characteristic of the organic component [136].

Polymer composites incorporating fillers such as calcium phosphate (CaP) or hydroxyapatite (HA) and other inorganic fillers have garnered significant interest for their versatile applications, particularly in the realms of biomedical and engineering fields (Table 1). Calcium phosphate, encompassing variations like hydroxyapatite, tricalcium phosphate and biphasic calcium phosphate, exhibits notable biocompatibility, rendering it suitable for applications in bone tissue engineering and dental materials. Similarly, hydroxyapatite, mimicking the natural mineral composition of bone, offers excellent biocompatibility and is commonly employed in orthopedic and dental implants. These fillers are seamlessly integrated into polymer matrices, including poly(lactic acid), poly(glycolic acid), and polyethylene glycol, using various processing techniques, such as melt-blending and electrospinning. The resulting composites exhibit enhanced mechanical properties, bioactivity and biocompatibility, making them advantageous for tissue engineering and medical implant applications. However, challenges such as achieving uniform filler dispersion and addressing processing compatibility need careful consideration in the design and fabrication of these advanced materials.

**Table 1.** List of inorganic fillers with their applications.

| Si. No. | Inorganic Fillers | Applications | Ref. |
|---|---|---|---|
| 1 | Alumina | Skin, orthopedic, drug delivery, dental implants, maxillofacial implants. | [137–141] |
| 2 | Tricalcium phosphate | Orthopedic, maxillofacial implants, dental implants, bone tissue engineering. | [138,142,143] |

**Table 1.** *Cont.*

| Si. No. | Inorganic Fillers | Applications | Ref. |
|---|---|---|---|
| 3 | Calcium phosphate | Dental implants, periodontal and alveolar bone regeneration, cranial bone defects, bone tissue engineering, cancer bone treatment. | [144–149] |
| 4 | Laponite nano clay | Drug delivery, wound dressing and healing, tissue engineering, biosensors. | [150–157] |
| 5 | Montmorillonite (MMT) | Drug release, bone regeneration, tissue engineering, periodontal regeneration, maxillofacial tumor treatment. | [158–168] |

## 6. Future Outlooks

The future outlook for natural polymers in bone tissue engineering involves addressing challenges in controlling the release of bioactive compounds, determining optimal concentrations for effective administration and overcoming limitations in scaffold fabrication. Advanced technologies, such as 3D bioprinting, offer a promising avenue for creating biomimetic scaffolds using a blend of natural polymers, bioactive molecules and living cells. Understanding the interactions between natural polymers and host tissues is crucial for progress in bone tissue regeneration. Research on the interaction of cells with natural biomaterials remains essential, given the difficulty in predicting the effects of extracellular substances. The adjustable and viscoelastic characteristics of natural biomaterials necessitate further exploration for the informed design of diverse scaffolding components. Additionally, the immunological response to scaffolds has gained recent attention, highlighting the pivotal role of the immune system in modulating tissue regeneration. Future studies will contribute to the development of advanced biomaterial production techniques. The emerging field of smart composite materials, characterized by the ability to sense and react to external stimuli, holds promise in diverse applications such as aircraft engineering, flexible electronics and medical devices. Shape memory effects, wherein materials return to their initial configuration in response to external inputs, are a key feature. Smart composites, derived from smart polymers, exhibit enhanced mechanical performance and can be powered by various sources, including electricity, light, heat and water moisture. Continued research is expected to lead to increasingly sophisticated biomaterial production techniques in this domain.

## 7. Conclusions

Natural polymers offer significant advantages compared to synthetic polymers. Their biocompatibility, biodegradability, low immunogenicity and non-toxic nature make them ideal candidates for applications in bone tissue engineering. The functionalization of natural polymers with bioactive compounds is crucial for enhancing bone regeneration. However, controlling the release of these bioactive elements and determining the optimal concentration for effective administration poses challenges. Traditional fabrication procedures make it difficult to produce scaffolds with adjustable dimensions and pore size. Advanced technology is required to process these natural polymers effectively. Bio-printing technologies offer a solution by enabling the fabrication of three-dimensional biomimetic scaffolds that blend natural polymers, bioactive molecules and living cells. A comprehensive understanding of the interactions between natural-based polymers and host tissues is essential to advance bone tissue regeneration and engineering. Despite significant efforts to comprehend cell–biomaterial interactions, predicting the impact of extracellular substances on cells remains challenging. Further research on the interactions of natural biomaterials is necessary to inform the design of versatile scaffolding components due to their adjustable and viscoelastic characteristics.

Recent attention has focused on the immunological response to scaffolds. Circulating white blood cells rapidly reach the implant site within hours to days, influencing the subsequent tissue regeneration process. Investigating the immune system's role in scaffold response is crucial for modulating future tissue regeneration. Ongoing research will contribute to the development of increasingly sophisticated biomaterial production

techniques. Smart composite materials possess the ability to sense changes in the external environment and adapt by modifying their qualities and geometric arrangements. Widely used in aircraft engineering, flexible electronics and medical devices, smart composites are derived from smart polymers, with enhanced mechanical performance. The shape memory effects of smart materials allow them to revert from a transient configuration to their initial configuration in response to external inputs. Smart composites can be powered by electricity, light, heat or water moisture, depending on the specific materials and structures employed in their creation.

**Author Contributions:** Conceptualization—A.M., Validation—A.K. and M.G.; Formal Analysis—A.M., Investigation—S.S. and P.S.; Resources—A.M., Writing—Original draft preparation—S.S., P.S. and Z.T., Writing—review and editing—A.M., A.K., M.G. and N.M., Visualization—A.M. and N.M., Supervision—M.G., N.M. and A.M.; and APC (Funding) Acquisition—N.M. All authors have read and agreed to the published version of the manuscript.

**Funding:** This review article received no external funding.

**Institutional Review Board Statement:** Not applicable.

**Informed Consent Statement:** Not applicable.

**Acknowledgments:** The authors are thankful to Lovely Professional University for providing a workplace for the completion of the review paper. Narsimha Mamidi acknowledges the University of Wisconsin-Madison, Madison, Wisconsin, and USA for the research facilities.

**Conflicts of Interest:** The authors have no conflict of interest.

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
