# Peer review of "Harnessing Natural Polymers for Nano-Scaffolds in Bone Tissue Engineering: A Comprehensive Overview of Bone Disease Treatment"

_cimb, doi:10.3390/cimb46010038_

Round 1

Reviewer 1 Report

Comments and Suggestions for Authors

This review-article addresses the Nano-scaffold Perspectives for the Treatment of Bone Diseases through Bone Tissue Engineering.

The article is well-written and covers many aspects regarding the bone loss. However, it would be great if the authors enrich the article with biomaterials that can be applied to more clinical topics.

In particular, they should include bone loss and regenerative medicine in the oral cavity (e.g. composite scaffolds and composite multifunctional barrier membranes). Moreover, the authors should address the importance of inorganic biomaterials in combination with polymers in more detail. They can also mention the importance of local drug delivery systems (use of mesoporous nanoparticles).

They can refer to the following articles:

10.3390/nano12050850

10.3390/pharmaceutics15010012

10.3390/molecules26030619

10.1007/s10965-022-02928-4

10.3390/pharmaceutics15030819

10.3390/ma13071748

They should mention the importance of the development of new smart composite materials (limitations of existing materials) their scientific and social impact.

Comments on the Quality of English Language

Minor editing of English language required

Author Response

Reviewer #1:

This review-article addresses the Nano-scaffold Perspectives for the Treatment of Bone Diseases through Bone Tissue Engineering.

  1. The article is well-written and covers many aspects regarding the bone loss. However, it would be great if the authors enrich the article with biomaterials that can be applied to more clinical topics. In particular, they should include bone loss and regenerative medicine in the oral cavity (e.g., composite scaffolds and composite multifunctional barrier membranes).

Response: We sincerely thank the reviewer for the helpful comment. As per the reviewer’s suggestion, we have included the bone loss and regenerative medicine in the oral cavity on page number 18 of the revised manuscript and highlighted in yellow color.

  1. The authors should address the importance of inorganic biomaterials in combination with polymers in more detail.

Response: We sincerely thank the reviewer for the helpful comment. As per the reviewer’s suggestion, the importance of inorganic biomaterials in combination with polymers is discussed on pages number 4 and 19 of the revised manuscript and highlighted in yellow color.

  1. They can also mention the importance of local drug delivery systems (use of mesoporous nanoparticles). They can refer to the following articles:

10.3390/nano12050850, 10.3390/pharmaceutics15010012, 10.3390/molecules26030619,

10.1007/s10965-022-02928-4, 10.3390/pharmaceutics15030819, 10.3390/ma13071748

Response: We sincerely thank the reviewer for the helpful comment. As per the reviewer’s suggestion, the importance of local drug delivery systems (mesoporous nanoparticles) is mentioned on page number 18 of the revised manuscript.

  1. They should mention the importance of the development of new smart composite materials (limitations of existing materials) their scientific and social impact.

Response: We sincerely thank the reviewer for the helpful comment. We added the importance of the development of new smart composite materials and their scientific and social impact on page number 20 of the revised manuscript and highlighted in yellow color.

English Language: We have thoroughly checked the English and corrected it throughout the revised manuscript.

Reviewer 2 Report

Comments and Suggestions for Authors

The review "Nano-scaffold Perspectives for Treatment of Bone Diseases
through Bone Tissue Engineering" reports on the application of different natural polymers in bone tissue engineering.

I appreciate the overall work of the authors to realize a comprehensive review on this subject, but I identified some weaknesses that need corrections.

First, the provenience of the Figures must be discussed, because these figures are not attributed to the authors. Even, these are modified, the scientific source must be cited in the legend.

In Section 2, the description of each polymer is too consistent and must be reduced. In this section the use of these materials is also discussed for cartilage tissue, which is different form the bone tissue engineering. So, I recommend that these references to be removed.

Table 1 has no relevance for the subject and must to be removed.

Author Response

The review "Nano-scaffold Perspectives for Treatment of Bone Diseases
through Bone Tissue Engineering" reports on the application of different natural polymers in bone tissue engineering.

I appreciate the overall work of the authors to realize a comprehensive review on this subject, but I identified some weaknesses that need corrections.

  1. First, the provenience of the Figures must be discussed, because these figures are not attributed to the authors. Even, these are modified, the scientific source must be cited in the legend.

Response: We sincerely thank the reviewer for the helpful comment. As per the reviewer’s suggestion, the scientific source is cited in the legend on page number 5 of the revised manuscript and highlighted in yellow color.

  1. In Section 2, the description of each polymer is too consistent and must be reduced. In this section, the use of these materials is also discussed for cartilage tissue, which is different form bone tissue engineering. So, I recommend that these references be removed.

Response: We sincerely thank the reviewer for the helpful comment. As per the reviewer’s suggestion, we have removed the cartilage tissue engineering and its references from the revised manuscript, and we have also reduced the content of each polymer.

  1. Table 1 has no relevance for the subject and must to be removed.

Response: We sincerely thank the reviewer for the helpful comment. As per the reviewer’s suggestion, we have also removed Table 1 from the revised manuscript.

Reviewer 3 Report

Comments and Suggestions for Authors

In this work authors presented knowledge on utilization of natural polymers in scaffold preparation, along with their applications in recent research trends and developments in the field of of bone tissue engineering. Some issues need to be considered:

-       Title: please add „natural polymers”

-       Authors focus on composite materials, but please mention hybrid materials in which there is a covalent bond between the components

-       What about the risks when using nano-structures, especially if their critical concentration is exceeded?

-       In nanotechnology we often observe agglomeration effects – how they influence on the key properties in bone tissue engineering?

-       Authors describe polymer composites with different fillers, such as calcium phosphate or hydroxyapatite – please provide a new table with the inorganic fillers used

-       What are the future outlooks in this area?

Author Response

Q1. Title: please add „natural polymers”

Response: We sincerely thank the reviewer for the helpful comment. As per the reviewer’s suggestion, we modified the title of the revised manuscript as follows:

“Harnessing Natural Polymers for Nano-Scaffolds in Bone Tissue Engineering: A Comprehensive Overview on Bone Disease Treatment”.

Q2. Authors focus on composite materials, but please mention hybrid materials in which there is a covalent bond between the components.

Response: We sincerely thank the reviewer for the helpful comment. As per the reviewer’s suggestion, we added the hybrid materials in which there is a covalent bond between the components on page number 19 of the revised manuscript and highlighted in yellow color.

Q3. What about the risks when using nano-structures, especially if their critical concentration is exceeded?

Response: We sincerely thank the reviewer for the helpful comment. Please see the following information to answer your question.

“The utilization of nanostructures across diverse domains, including medicine, electronics, and materials science, holds significant potential, albeit accompanied by inherent risks and challenges. An imperative facet involves comprehending and addressing the potential hazards linked to nanostructure deployment, particularly when surpassing critical concentrations. Notably, the concept of toxicity emerges as a paramount consideration, given that nanostructures may manifest distinct toxicological traits compared to their bulk counterparts. The likelihood of toxicity escalates upon exceeding critical concentrations, necessitating a thorough evaluation of potential adverse impacts on human health and the environment. Additionally, the intricate interplay between nanostructures and biological systems, attributed to their diminutive size and expansive surface area, introduces a layer of complexity. Surpassing critical concentrations amplifies these interactions, potentially leading to unforeseen biological responses with harmful effects. Environmental ramifications are also a concern, as the deliberate or inadvertent release of nanostructures may disrupt ecosystems by contributing to the accumulation of nanoparticles. The understanding of the long-term effects of nanostructure exposure remains incomplete, emphasizing the need for investigations into cumulative impacts, chronic toxicity, and broader environmental consequences associated with exceeding critical concentrations. Furthermore, the tendency of nanostructures to agglomerate or aggregate could be exacerbated beyond critical concentrations, influencing their stability and reactivity. Robust risk assessment strategies, encompassing exposure pathways, toxicity profiles, and application-specific risks, are crucial for ensuring the safety of nanostructures. A regulatory framework is essential to guide responsible development and usage, demanding regulatory agencies stay abreast of nanotechnological advancements and establish standards for safe practices. Researchers and industries engaged in nanostructure applications must adopt precautionary measures, conduct meticulous risk assessments, and adhere to safety guidelines to mitigate potential risks. Continuous research efforts are indispensable to augment our understanding of nanostructure behavior and effects, especially given the relentless progression of technology.”

Q4. In nanotechnology we often observe agglomeration effects – how they influence on the key properties in bone tissue engineering?

Response: We sincerely thank the reviewer for the helpful comment. We believe that the following information will clarify your question.

“Agglomeration effects play a pivotal role in the realm of nanotechnology, particularly within the context of bone tissue engineering, exerting a substantial influence on key properties and outcomes. Various facets underscore the impact of agglomeration on the performance of nanomaterials in this field. Firstly, in terms of biocompatibility, agglomerated nanoparticles can modify the surface properties of nanomaterials, potentially affecting interactions with surrounding cells and tissues and, consequently, the success of implant integration and tissue regeneration. Moreover, cellular uptake by bone cells may be influenced, as larger agglomerates exhibit reduced internalization compared to individual nanoparticles, potentially compromising the cellular response crucial for tissue regeneration. The critical factor of surface area and reactivity is also affected, as agglomeration diminishes the effective surface area available for interactions, impacting the release of bioactive molecules and the stimulation of specific cellular responses. Additionally, the mechanical properties of nanocomposites used in bone tissue engineering, such as scaffolds or implants, can be compromised by agglomeration, leading to variations in mechanical strength and stiffness. The use of nanoparticles for drug delivery in bone tissue engineering is not immune to agglomeration's influence, potentially causing non-uniform drug distribution and suboptimal therapeutic effects. Agglomerated nanoparticles may induce different inflammatory responses compared to dispersed ones, potentially leading to adverse reactions in the surrounding tissue. In animal studies or clinical applications, the in vivo behavior of agglomerated nanoparticles may differ, impacting the overall efficacy and safety of bone tissue engineering approaches. Researchers in bone tissue engineering diligently employ strategies such as surface modifications, dispersing agents, and fabrication technique optimization to address and control agglomeration effects, thereby enhancing the performance of nanomaterials for effective and safe bone regeneration applications.”

Q5. Authors describe polymer composites with different fillers, such as calcium phosphate or hydroxyapatite – please provide a new table with the inorganic fillers used

Response: We sincerely thank the reviewer for the helpful comment. As per the reviewer’s suggestion, we have added Table 1 on page number 20 of the revised manuscript and highlighted it in yellow color.

In addition, we added the following information on the same page of the revised manuscript.

“Polymer composites incorporating fillers such as calcium phosphate (CaP) or hydroxyapatite (HA) and other inorganic fillers have garnered significant interest for their versatile applications, particularly in the realms of biomedical and engineering fields. Calcium phosphate, encompassing variations like hydroxyapatite, tricalcium phosphate, and biphasic calcium phosphate, exhibits notable biocompatibility, rendering it suitable for applications in bone tissue engineering and dental materials. Similarly, hydroxyapatite, mimicking the natural mineral composition of bone, offers excellent biocompatibility and is commonly employed in orthopedic and dental implants. These fillers are seamlessly integrated into polymer matrices, including poly(lactic acid), poly(glycolic acid), and polyethylene glycol, using various processing techniques such as melt blending and electrospinning. The resulting composites exhibit enhanced mechanical properties, bioactivity, and biocompatibility, making them advantageous for tissue engineering and medical implant applications. However, challenges such as achieving uniform filler dispersion and addressing processing compatibility need careful consideration in the design and fabrication of these advanced materials.”

Q6. What are the future outlooks in this area?

Response: We sincerely thank the reviewer for the helpful comment. As per the reviewer’s suggestion, the following future outlook was added to the revised manuscript.

“The future outlook for natural polymers in bone tissue engineering involves addressing challenges in controlling the release of bioactive compounds, determining optimal concentrations for effective administration, and overcoming limitations in scaffold fabrication. Advanced technologies, such as 3D bioprinting, offer a promising avenue for creating biomimetic scaffolds using a blend of natural polymers, bioactive molecules, and living cells. Understanding the interactions between natural polymers and host tissues is crucial for progress in bone tissue regeneration. Research on the interaction of cells with natural biomaterials remains essential, given the difficulty in predicting the effects of extracellular substances. The adjustable and viscoelastic characteristics of natural biomaterials necessitate further exploration for the informed design of diverse scaffolding components. Additionally, the immunological response to scaffolds has gained recent attention, highlighting the pivotal role of the immune system in modulating tissue regeneration. Future studies will contribute to the development of advanced biomaterials production techniques.

The emerging field of smart composite materials, characterized by the ability to sense and react to external stimuli, holds promise in diverse applications such as aircraft engineering, flexible electronics, and medical devices. Shape memory effects, wherein materials return to their initial configuration in response to external inputs, are a key feature. Smart composites, derived from smart polymers, exhibit enhanced mechanical performance and can be powered by various sources, including electricity, light, heat, and water moisture. Continued research is expected to lead to increasingly sophisticated biomaterial production techniques in this domain.”

Round 2

Reviewer 3 Report

Comments and Suggestions for Authors

Authors provided proper explanations  to the reviewer's comments, and the revised manuscript can be published.